# Physiotherapy under pressure: A cross-sectional study on the interplay between perfectionism, moral injury, and burnout

**Daniel Biggs**[1,2], **Laura Blackburn**[2], **Cameron Black**[2,3], **Sivaramkumar Shanmugam**[2]*

1 Academy of Sport and Wellbeing, University of Highlands and Islands, UHI Perth, Perth, United Kingdom, 2 Department of Physiotherapy and Paramedicine, School of Health and Life Sciences, Glasgow Caledonian University, Glasgow, United Kingdom, 3 Division of Occupational Health, Buckinghamshire Healthcare NHS Trust, Aylesbury, United Kingdom

* sivaram.shanmugam@gcu.ac.uk

## Abstract

### Background

Given the escalating challenges for UK-based physiotherapists in workload pressures, budget constraints, staff shortages and patient wait times, the profession (of 65,000 registered physiotherapists) necessitates immediate attention to the health and well-being of the therapists. This pioneering study aims to examine perfectionism, moral injury, and burnout among UK-based physiotherapists across the NHS, private practice, sports, and academia.

### Method

This cross-sectional study utilised an online survey and implementation of Structure Equation Modelling (SEM) to assess the interplay of perfectionism (Multidimensional Perfectionism Scale-Short Form), moral injury (Moral Injury Symptoms Scale-Healthcare Professionals), and burnout (Shirom-Melamed Burnout Questionnaire). Our sample size calculation represents the UK physiotherapy profession, utilising a 95% confidence interval with a 5% margin of error.

### Findings

Our analysis conducted on (n = 402) UK-based physiotherapists reveals significant burnout levels, with 96% of participants presenting with moderate to high burnout scores. SEM revealed perfectionism and moral injury collectively accounted for a substantial 62% of burnout variability, highlighting their sequential impact on burnout manifestation.

### Interpretation

With such high levels of burnout, urgent intervention is paramount. Elevated burnout presents challenges for the physiotherapy profession as staff retention, accurate and effective patient care, and overall health are severely impacted due to burnout. Recognising and addressing perfectionism and moral injury, such as through amendment or development of policy, becomes pivotal to mitigate its impact on individual and collective health.

**Data Availability Statement:** Data are available from the Glasgow Caledonian University Institutional Data Access / Ethics Committee at HLSEthicsPSWAH@gcu.ac.uk. (or contact via the

primary author – Dr Sivaramkumar Shanmugam). Based on the University's process, data will be stored on password protected devices for 10 years and will be destroyed confidentially. The study complies with the Data Protection Act (2018) and the General Data Protection Regulation (GDPR). The data controller is Glasgow Caledonian University. Information is being processed on the basis of Article 6(1)(e) of the General Data Protection Regulation and to perform a task carried out in the public interest. Enquiries specifically relating to data protection should be made to the University's Data Protection Officer (DPO). The DPO can be contacted by email: dataprotection@gcu.ac.uk. Following the 10 year period, raw data will be deleted, however, anonymised data for the paper is available as a supplementary file.

**Funding:** The author(s) received no specific funding for this work.

**Competing interests:** No competing interests.

# Introduction

## Perfectionism

Bayeux Tapestry encapsulates the essence of perfectionism through its intricate embroidery process, where each thread is meticulously woven to depict mediaeval scenes with precision. Embroiderers become immersed in a battle, fixated on capturing every detail to ensure the tapestry emerges as a flawless masterpiece. Parallels between Bayeux Tapestry and perfectionism extends beyond the process to encompass the high standards of the creators. In both cases, there is commitment to achieving absolute flawlessness, be it the careful stitching of threads or the pursuits of perfection in one's work. Bayeux Tapestry thus serves as a tangible representation of the relentless pursuit of flawlessness, mirroring the psychology of perfectionism. Like the embroiderers of the tapestry, modern professionals, such as physiotherapists often strive for similar levels of flawlessness in their work.

Perfectionism is commonly acknowledged as an intricate multifaceted personality trait and often be characterised by exceedingly high levels of self-critical judgement and personal standards, encompassing a synthesis of rigorous self-evaluative standards [1]. Frost et al. [2] outlined six dimensions of perfectionism. These encompassed Personal Standards, characterised by high internal expectations; Concerns Over Mistakes, reflecting apprehensions about errors; Parental Criticism, entailing critical evaluations/input from parents; Parental Expectations, involving demands of perceived flawlessness form parents; Doubts About Actions, encapsulating subjective uncertainty about performance; and Organisation, emphasising preparedness and methodicalness. The framework explores elevated self expectations, maladaptive cognitive patterns related to mistakes, inward perfectionism, perceived parental pressure for flawlessness, subjective uncertainty about performance, and the acknowledgement of preparedness and methodical approaches. In contrast, Hewitt and Flett's model [3] explains perfectionism as multidimensional and simplifies it into 3 dimensions. These dimensions are Self Oriented Perfectionism (SOP), characterised by an inherent drive for flawlessness leading to self-criticism and high personal standards; Socially Prescribed Perfectionism (SPP), arising from external expectations form significant others like parents, guardians, coaches or teachers, resulting in excessive self-criticism when unmet; and Other Oriented Perfectionism (OOP), revolving around expecting perfection from others and manifesting in hyper-critical behaviour. The current study's interests are in understanding the relationship between the expectations placed on the individual based on self set standards or perceived expectations placed on them by others and how this interacts with rates of moral injury and burnout. Therefore, it is the perspective of the authors that investigating OOP would not add additional insight in understanding this relationship.

Despite disparity in dimensions, common themes resonate between the two models, such as the imposition of high standards in both internal and external contexts and the presence of maladaptive cognitions during the pursuit and outcomes of goals. Literature favours Hewitt and Flett's model, drawn by its validation through factor analysis, supporting the existence of adaptive and maladaptive facets of perfectionism [4].

A recent review focused on SPP, describing it as an epidemic [5]. Flett et al. [5] showed the destructive nature of SPP through links with poor mental well-being, physical health, interpersonal adjustment, suicidal cognitions and tendencies, and burnout. The review stated, "SPP is a significant public health concern that urgently requires sustained prevention and intervention efforts" [5(p1)]. As SPP is a pressure to meet external standards prescribed by the social environment, it may be problematic in certain settings and job roles, particularly when multiple factors contribute to decision-making. In the realm of healthcare, UK-based physiotherapists operate as independent practitioners across various sectors, such as public, private or

sports settings. It is noteworthy that a significant majority, approximately 70% of UK-based physiotherapists work in the NHS [6]. While these practitioners enjoy autonomy in their roles, it is important to recognise their independence is circumscribed by NHS procedures. Like the Bayeux Tapestry, physiotherapists must carefully consider numerous factors, meticulously weaving together individual patient needs with the intricate patterns of organisational demands. They are tasked with harmonising these diverse needs and standards, striving for a tapestry of care that is both comprehensive and precise. The pressure of multiple expectations partnered with the nature of perfectionism can become problematic in decision making and often cause internal conflict in healthcare. However, this intricate balancing act, much like the creation of the tapestry, may introduce elements of unpredictability, potentially leading to discrepancies among the therapist, patient, and the organization, which could ultimately compromise the practitioner's ability to provide optimal care.

## Moral injury

Moral injury, rooted in the ethical branch of axiology and philosophy, revolves around conflict between an individual's moral beliefs and the actions they witness and partake in [7]. This form of injury often occurs when there is a perceived betrayal of moral integrity, from either someone of legitimate authority or oneself during a high-stakes situation [8]. Moral injury renders feelings of shame, remorse, meaninglessness, and grief due to the violation of core moral beliefs [9]. A violation of a moral beliefs can significantly alter personal identity.

A growing body of literature explores moral injury in healthcare professionals. The oath taken by healthcare practitioners, prioritising patient needs, serves as a guiding principle for their conduct and decisions [10]. However, competing factors such as insurance considerations, hospital dynamics, electronic medical records, the healthcare system model, and personal financial security can take precedence over patient well-being and potentially compromise their care. Over time, decisions deviating from the best interests of patients can be perceived as a moral injustice, leading to feelings of shame and guilt. These emotions, in turn, may contribute to a desire to leave the healthcare profession [11]. This mirroring the concept of SPP, where the pursuit of flawless decisions and standards may be hindered by external factors, making the attainment of perfection seemingly unachievable.

Čartolovni et al. [12] builds on our understanding of moral injury [10] by highlighting how conflict between a practitioner's duty to provide care, their fundamental professional role, and the witnessing of traumatising events impose an ethical burden on them. This significantly impacts the mental health of healthcare professionals. Instances of trauma include the inadequate allocation of resources, a pervasive sense of hopelessness and helplessness in overcrowded hospitals, and the emotional toll of witnessing innocent deaths, resulting from moral decision-making in triage prioritisation [12]. This underscores that moral injury is not a uniform experience; rather, healthcare professionals face vulnerability arising from various aspects of their roles in the job.

Moral injury must be clearly distinguished from Post-Traumatic Stress Disorder (PTSD), this differentiation is crucial in the healthcare context, where the focus lies on understanding how moral injury arises, from the frustration experienced by healthcare professionals. Although these individuals are trained to provide care, they encounter barriers leading to an inability to deliver the expected care. This discrepancy can result in various adverse psychological outcomes, including burnout. Notably, research has demonstrated a correlation between burnout and moral injury [13]. Recognising and appreciating these distinctions is important for healthcare practitioners and researchers, as it sheds light on the unique challenges posed by moral injury, and the resultant repercussions in healthcare settings.

## Burnout

A syndrome with far-reaching consequences, burnout affects not only the well-being and performance of individuals but also organisational outcomes, such as employee turnover, absenteeism, and diminished productivity [14]. Three core dimensions of exhaustion, cynicism, and loss of commitment albeit with subtle changes are conceptualised by Maslach and Jackson [15]. These authors show burnout as a tri-dimensional syndrome, encompassing Emotional and Psychophysical Exhaustion, linked to somatic-like symptoms, such as tiredness or headaches; Depersonalisation, involving the tendency to detach oneself from society and societal values; and Reduced Sense of Accomplishment, indicating an adverse appraisal of individual abilities or achievements. Since the inception of this model, various others have been developed and grounded in the three dimensions of Maslach and Jackson's model [15].

The Shirom-Melamed Burnout Measure (SMBM) has been validated through confirmatory factor analysis [16]. SMBM is characterised by three dimensions: emotional exhaustion (EE), physical fatigue (PF) and cognitive weariness (CW). Burnout onset is influenced by a range of factors at the individual (micro), occupational (meso), and organisational (macro) levels. These factors, encompassing high job demands, insufficient resources and support, a lack of autonomy and control, role ambiguity, and interpersonal conflicts are further influenced by the interaction of personality traits, coping styles, and prior experiences, shaping the emergence and progression of burnout [17].

Donohoe et al. [18] revealed 40% experienced moderate-to-high levels of burnout in physiotherapists. Subsequent studies consistently delved into the phenomenon of burnout in physiotherapists, reinforcing the vulnerability of physiotherapists to burnout [19]. It is important to note these studies span various countries with different healthcare systems, and the impact of contributing factors may vary across contexts. Nevertheless, the collective evidence indicates burnout is a prevalent and significant issue among physiotherapists, reaching moderate-to-high levels across diverse settings.

Literature on burnout in physiotherapists spans different countries and continues to centre on prevalence. However, there is a notable absence of research on burnout in UK physiotherapists. Investigating this aspect is important for advancing our understanding, allowing for international and interdisciplinary comparisons, and the development of appropriate and specialist interventions. It also provides an opportunity to examine burnout within the context of the UK healthcare system, particularly post-pandemic.

## Perfectionism, moral injury and burnout

Research highlights links between perfectionism, moral injury, and burnout [13, 20]. For example, Testoni et al. [13] found a correlation (r .20 p = < .001) between moral injury and burnout in clinicians. This finding is important as clinicians tend to dehumanise patients and colleagues during high levels of burnout, when high levels of exposure to moral injury occurred [ibid].

Martin et al. [20] found perfectionism to significantly predict emotional exhaustion (β = 0.55, p = < .001) and depersonalisation (β = 0.18, p = .006) in physicians, while Biggs, McKay and Shanmugam [21] showed SPP to be associated with burnout (β = 0.36 p = .001) in physiotherapy students, suggesting perfectionism renders them vulnerable to burnout. Against this backdrop, we argue the tendency to achieve perfection in the workplace adversely affects performance, placing individuals at risk of burnout.

The intersection between perfectionism and moral injury is of particular interest, especially in how SPP might interact with moral injury. One plausible theorised outcome is SPP could lead to feelings of betrayal, subsequently triggering moral injury. To the authors' knowledge,

perfectionism and moral injury have not been investigated together. However, it seems logical SPP could serve as a predictor of moral injury. The relationship between perfectionism, burnout and moral injury has also not been explored. Given the established associations of both perfectionism and moral injury with burnout in the literature [13, 20], and considering perfectionism predicts moral injury, it is reasonable to infer a connection between these variables. These three variables, despite limited research, each demonstrated statistical significance in healthcare professionals worldwide, with no data, however, pertaining to perfectionism, moral injury and burnout in UK-based physiotherapists. This study aims to be the first to comprehensively examine these three variables among UK-based physiotherapists. We anticipate, firstly, a significant association between perfectionism and moral injury with burnout. Secondly, perfectionism, we expect, will predict moral injury, and in turn burnout. Therefore, this investigation aims to understand the nature and extent of the relationship between perfectionism, moral injury and burnout in UK-based Physiotherapists.

## Method

### Participants

Between October 2022 and February 2023, 402 UK-based physiotherapists completed an online survey. Participants were categorised as: Private Physiotherapists (9%), NHS Public Physiotherapists (87%), Sports (<1%), Academia (3%) and Other (<1%) (see Table 1). All worked in a range of clinical and non-clinical physiotherapy environments and specialties, including inpatient, outpatient, and higher education. All reported different duration of employment, with 23% working 2–5 years, 21% over 10 years, and 19% 12–24 months in their current position. See Table 2 for an overview of demographics, including gender, ethnicity, and education. Recruitment occurred through convenience sampling, advertising through social media and through emails with an invitation to complete the survey.

### Methodology

In this study, we followed the "Strengthening the Reporting of Observational Studies in Epidemiology" (STROBE) guidelines. Our methodology was specifically utilised the cross-sectional study version of the STROBE Checklist, incorporating 22 key points for survey design and result reporting, as detailed in S1 Table.

**Table 1. Categorisation of physiotherapy positions.**

| Categorisation of Participants | Number within the field | Percentage |
|---|---|---|
| Band 5 NHS | 52 | 13 |
| Band 6 NHS | 91 | 23 |
| Band 7 NHS | 129 | 32 |
| Band 8a | 55 | 14 |
| Band 8b | 17 | 4 |
| Band 8c | 3 | <1 |
| Band 8d | 1 | <1 |
| Private Practice | 35 | 9 |
| Sports-Based | 3 | <1 |
| Lecturer | 5 | 1 |
| Senior Lecturer | 7 | 2 |
| Reader | 1 | <1 |
| Other | 3 | <1 |

**Table 2. Frequency and percentage of the key demographics.**

| Demographics | |
|---|---|
| **Gender** | |
| Female | 330 (82%) |
| Male | 67 (17%) |
| Non-binary | 1 (<1%) |
| Prefer not to say | 4 (1%) |
| **Age** | |
| 21–24 years | 29 (7%) |
| 25–29 years | 40 (10%) |
| 30–34 years | 52 (13%) |
| 35–39 years | 71 (18%) |
| 40–44 years | 77 (19%) |
| 45–49 years | 55 (14%) |
| 50+ years | 78 (19%) |
| **Ethnicity** | |
| Asian or Asian British | 28 (7%) |
| Black, Black British, Caribbean or African | 8 (2%) |
| Mixed or Multiple Ethnic Groups | 18 (4%) |
| White | 348 (87%) |
| **Education** | |
| BSc | 191 (48%) |
| MSc | 164 (41%) |
| PhD | 22 (5%) |
| Professional Doctorate | 1 (<1%) |
| DPT | 8 (2%) |
| PgDip | 13 (3%) |
| MRes | 3 (1%) |

Bachelor of Science. MSc = Master of Science. PhD = Doctor of Philosophy. DPT = Doctor of Physiotherapy.

PgDip = Postgraduate Diploma. MRes = Master of Research.

## Measures

Participants completed a Microsoft (MS) Forms survey which consisted of 12 questions related to demographics and work environment, alongside three validated scales, measuring perfectionism, moral injury, and burnout in the profession. Perfectionism was measured via the Multidimensional Perfectionism Scale Short Form [4] (MPS-SF), which measures the aforementioned dimensions of self oriented perfectionism (SOP) and socially prescribed perfectionism (SPP). Moral Injury was measured via Moral Injury Symptoms Scale- Healthcare Professionals Version (MISS-HF) [22]. Due to limitations on the platform hosting the survey, the original Likert scale used to measure moral injury required modification from 10 levels to 7. Finally, Burnout was measured via the Shirom-Melamed Burnout Questionnaire [23] (SMBM) using the domains of emotional exhaustion (EE), physical fatigue (PF) and cognitive weariness (CW). The Likert scale used to measure burnout also required modification to accommodate the survey platform from a Likert of 7 to 5.

## Procedure

Participants found the survey through links on posts from the investigators' social media posts and email campaigns. By clicking on the link, participants are directed to the MS Forms

platform where they can complete the questions. Inclusion criteria included participants 21 years of age or over and work as a physiotherapist in the United Kingdom.

## Ethics

Informed consent was provided by selecting the 'yes' option when asked for consent to participate in the survey. The survey explicitly stated the purpose of the study, assured anonymity, confidentiality, and emphasised the voluntary nature of participation. Further information was provided in a participant information sheet. In the first question, participants were informed that by proceeding with the survey, they were providing consent to utilise their responses for research purposes. Participants could withdraw at any time without penalty. The start date of recruitment was 25 November 2022 and the end date 30 April 2023. They were informed of contact details for the research team and potential emotional impact. Post-survey, general resources for mental health support were provided. Ethical approval was obtained from Glasgow Caledonian University's School of Health and Life Sciences (HLS/PSWAHS/22/017).

## Data analysis and study size

Preliminary, descriptive, and correlation analysis were conducted in SPSS 26.0. Scores were categorised as low, medium, and high by dividing the highest possible score for each measure into three parts and used third quartile percentages to calculate the three categories. Preliminary analysis comprised out-of-range values, univariate, and multivariate normality, and reliability. Two-step structural equation modelling (SEM) was undertaken in AMOS 26.0 [24]. Maximum likelihood estimation was employed to assess the goodness-of-fit and parameters of the statistical model. The model included four interrelated latent variables: SSP, SOP, moral injury, and burnout. Burnout was indicated by three dimensions (EE, PF, and CW) and moral injury, SOP and SPP were indicated by their respective items. This approach enabled the power analysis calculation of the model that showed 95% confidence interval with a 5% margin of error within the sample and shows representation of the UK Physiotherapist population. A two step Structure Equation Model (SEM) evaluated model fit, starting with measurement model evaluation, followed by confirmatory factor analyses. Subsequently, the theorised structural relationships were evaluated. Conventional criteria of approximate markers of acceptance ($\chi^2$/df ratio < 3.00, IFI and CFI > 0.90, RMSEA < 0.08) and excellent fit ($\chi^2$/df ratio < 2.00, IFI and CFI > 0.95, RMSEA < 0.06) as recommended by Marsh et al. [25].

## Results

### Preliminary analyses, descriptive statistics, and bivariate correlations

Descriptive statistics, including the average (mean) and variance (standard deviation), correlations (Pearsons), and the level of significance for each total of the MPS-SF (excluding other oriented perfectionism), MISS-HF, and SMBQ domains are presented in Table 3. For the SOP and SPP domains of the MPS-SF, the highest 270 total score possible is 35. The maximum total score possible for the MISS-HF is 70. SMBQ highest possible total score is 70 while the highest scores for each domain include 42 for PE, 21 for EE, and 35 for CW. All values lay within the expected range of the scale. No dropouts or missing data occurred to the requirement in the survey for participants to complete each question. Data for all scales fit normal distribution models of absolute skewness and absolute kurtosis.

**Table 3. Mean, standard deviation, and correlations of the three outcome measures.**

| 1.SOP | 2.SPP | 3.Moral Injury | 4.Physical Exhaustion | 5.Emotional Exhaustion | 6.Cognitive Weariness | 7.Total Burnout | Mean | SD |
|---|---|---|---|---|---|---|---|---|
| 1 | .576** | .291** | .535** | .124* | .240* | .108* | 25.2 | 6.8 |
| | 2 | .482** | .540** | .346** | .397** | .336** | 22.4 | 6.7 |
| 3 | | | .504** | .461** | .432** | .559** | 34.7 | 8.4 |
| 4 | | | | .473** | .641** | .886** | 21.5 | 5.8 |
| 5 | | | | | .514** | .727** | 7.3 | 7.3 |
| 6 | | | | | | .873** | 15.2 | 5 |
| 7 | | | | | | | 44.1 | 12 |

*Note*. Cronbach Alpha—SOP .84, SPP .90, Moral Injury .74, Burnout .94, Sig level

* .05

** .01. SOP = socially oriented perfectionism. SPP = socially prescribed perfectionism.

All correlations between the total and domain total scores for each scale were found to be significant (see Table 3). Table 4 highlights the percentages of the sample by category level in outcome scores. As can be seen, 95% of physiotherapists experienced moderate-to-high levels of SOP, 92% had moderate-to-high levels of SPP, 90% had moderate-to-high levels of moral injury, and 96% had moderate-to-high levels of burnout.

## Structural equation modelling

Confirmatory factor analyses indicated acceptable-to-excellent fit for the measurement model ($\chi$2/df ratio = 2.25, IFI = 0.92, CFI = 0.92, RMSEA = 0.06 CI 0.06–0.07). Composite reliabilities ($\rho$c) supported the measurement model: SOP = 0.90; SPP = 0.84; Moral Injury = 0.74 and Burnout = 0.94. SEM also indicated an acceptable-to-excellent fit ($\chi$2/df ratio = 2.25, IFI = 0.92, CFI = 0.92, RMSEA = 0.06 CI 90% 0.06–0.07, SRMR = .067). All residual covariances were < 2.0. Overall, the model explained 62% variance in burnout. Parameters, including SPP, SOP, moral injury, and burnout, are displayed in Fig 1.

## Discussion

Our study sought to explore the relationship between perfectionism, moral injury, and burnout in physiotherapists based in the UK, through both correlation analyses and SEM techniques. We found a substantial proportion of variability in burnout can be accounted for by perfectionism and moral injury. As expected, we uncovered noteworthy positive relationships between SPP-Burnout, SPP-Moral Injury, and Moral Injury-Burnout. Notably, a statistically significant pathway emerged from perfectionism (SPP) to moral injury, and to burnout. This underscores two crucial findings. Firstly, SPP and moral injury exhibited meaningful associations with burnout. Secondly, SPP served as a precursor with moral injury, which in turn predicted burnout.

In essence, our study not only establishes the existence of significant relationships among SPP, moral injury, and burnout but also shows a sequential pattern wherein SPP sets the stage for moral injury, ultimately contributing to a maladaptive manifestation of burnout.

**Table 4. Population percentage by category level proportion in each outcome measure.**

| Level | Self-oriented Perfectionism | Socially Perfection | Prescribed | Moral Injury | Burnout |
|---|---|---|---|---|---|
| Low | 5% | 8% | | 10% | 4% |
| Moderate | 33% | 54% | | 84% | 56% |
| High | 62% | 38% | | 6% | 40% |

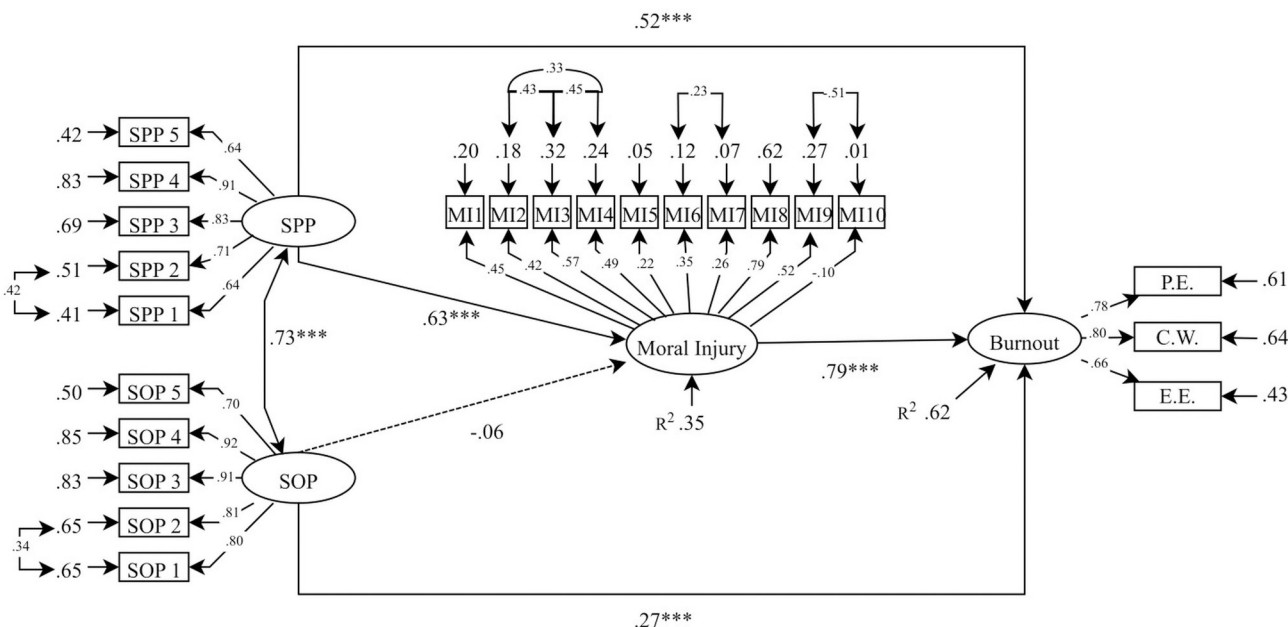

**Fig 1. Structural equation model to show the relationship between perfectionism, moral injury and burnout.**

## Burnout in physiotherapy

96% of physiotherapists in our sample suffered from moderate-to-high levels of burnout, presenting a pervasive challenge. Implications of these scores are concerning for several reasons. When considering our study and the research by Donohoe et al. [18] who found 40% of physiotherapists suffered moderate to high burnout levels, our burnout levels reveal a marked increase. These heightened levels present immediate challenges, with potential depletion of the physiotherapy workforce intensifying pressure on existing staff. Secondly, burnout has been linked to higher turnover rates among staff [26]. Given the existing shortage of physiotherapists in the UK, the high levels of burnout in this study may be a contributing factor to a significant exodus from the profession [27]. Our findings not only underscore the severity of burnout-related issues but also offer a potential explanation for the scarcity of physiotherapists in the UK.

Individuals can encounter challenges, such as impaired executive function, compromised episodic and working memory, and diminished task attention when facing burnout [28]. With appropriate intervention, these issues can be reversed [29]. Our examination of burnout dimensions reveals both physical and emotional exhaustion scores on average fall between medium and high. In a study by Brand et al. [30], 21.1% of health professionals with high burnout scores experienced symptoms such as difficulty falling or staying asleep, characteristic of insomnia. Given sleep deprivation adversely affects attention, working memory, long-term memory, and decision-making [31], it may explain why these symptoms have been reported as risk factors in burnout research [29]. It is then logical to infer insomnia might be prevalent among UK physiotherapists, given the burnout scores observed in our sample. As a result, despite experiencing exhaustion, achieving restful sleep may be elusive due to the coexistence of insomnia. If physiotherapists are indeed dealing with insomnia, the quality of care provided may be compromised, posing additional health risks to their patients.

Given that this study represents one of the first to capture burnout in a sample of UK physiotherapists, with such stark results, urgent and comprehensive efforts are required to mitigate

the impending risks to staff, patients, and the broader society. Understanding the reasons behind the increase in burnout rates requires further exploration.

## Perfectionism and burnout

On average, we observed moderate to high levels of SOP and SPP in our physiotherapy population. Correlational analysis reveals a connection between SPP, SOP, and moral injury and burnout. Further analysis indicates a positive association between both SPP and moral injury with burnout. The self-induced external perfectionistic standards appear to be the closest predictor of burnout. In practical terms, this connection could manifest when a physiotherapist achieves a set goal, but external hierarchical peers may fail to provide sufficient feedback that meets the maladaptive perfectionistic standards of the feedback-reciprocal physiotherapist. Consequently, the physiotherapist, in such cases, becomes vulnerable to experiencing elevated levels of burnout.

Our findings align with previous research identifying SPP as a stronger predictor of burnout compared to SOP [21]. Recently, Biggs McKay and Shanmugam [21] found a significant association between SPP and burnout in UK physiotherapy students, suggesting a consistent trend across different professional stages. Our findings also contribute to a growing body of evidence linking perfectionism, particularly SPP, to adverse health implications. SPP demonstrates a stronger direct relationship with clinically diagnosed conditions, including affective disorders, eating disorders, and narcissistic personality disorder [5]. The relationship between SPP and burnout observed in our sample underscores the importance of recognising and addressing maladaptive perfectionism tendencies, as they can have significant implications for individual health and well-being.

## Moral injury and burnout

Mantri et al. [22], in their sample of 1831 health professionals, found a significant relationship between moral injury and burnout. The authors understood the surge in rates of moral injury found in their data to be in response to the pandemic. On average, our sample consisted of moderate levels of moral injury. Further analysis revealed a significant correlation between burnout and moral injury, similar to previous research [13] and the understanding that burnout might not be due to pandemic related factors alone [22]. SEM helped us establish a significant association, with moral injury predicting burnout (β 0.78). Our study stands as the first to uncover such an association in the physiotherapy profession. Building on the work of Mantri et al. [22], we found physiotherapists prone to maladaptive perfectionism, who then experience a moral injury, have a higher risk of burnout.

Our study is the first to establish a relationship between SPP and moral injury (β 0.54). We theorise the external standards set by significant others could trigger a sense of betrayal, subsequently leading to moral injury. Our results support this association. Considering the mean SPP score in our sample is moderate to high, it implies that physiotherapists are striving for standards of perfection as influenced by their peers. However, this pursuit may render them susceptible to moral injury if perceived betrayal from significant peers, such as line managers or superiors, occurs. This vulnerability likely predisposes physiotherapists to a risk of elevated levels of burnout.

## Perfectionism, moral injury and burnout's pathway

This study marks the first instance where the relationship between perfectionism and burnout is elucidated from the perspective of a betrayal of values. The SEM results extend the findings of Martin et al. [20] and Testoni et al. [13], underscoring the susceptibility to burnout

exhibited by SPP physiotherapists when confronted with moral injury. SPP was not the only domain on the MPSSF to have a relationship with burnout. A relationship between SOP and burnout, observed through SEM, was not significantly associated by moral injury, despite a modest association between the two (β 0.27). Our research suggests SOP has a detrimental relationship with burnout, explaining some of the variance in scores. Similar to the weaving of the Bayeux Tapestry, SOP and burnout together can leave healthcare professionals vulnerable to heightened exhaustion and cognitive weariness as they strive for perfection.

## Practical implications

Our research underscores the importance of the interplay between perfectionism and moral injury for physiotherapists and their line managers. Although our research focuses on a physiotherapy population, high burnout levels are likely not unique to the profession and it can be expected to be found across other health professionals. Burnout, as such, can be considered a 'wicked' problem and of great concern to the sustainability of the NHS workforce. Challenges such as workload pressures, budget constraints, staff shortages and patient wait times would be and are increasing due to the sustainability of the practitioners in physiotherapy. Strategies need to be developed to manage risk factors for burnout and support those impacted by the syndrome at a national level. Creating these strategies will likely benefit from a collaborative effort between key stakeholders interested in supporting the mental health and job satisfaction of health care workers, including the healthcare workers affected. For example, the use of co-creation methodologies may facilitate interventions that are realistic, person-centred, and achievable. Stakeholders like the professional body, the Chartered Society of Physiotherapy (CSP), play a vital role in supporting physiotherapists and offer resources, training, and advocacy on issues such as workplace wellbeing, which includes burnout prevention and management.

## Limitations and future research directions

Although our study can explain around 60% of the variance of burnout in physiotherapists, other influences of the remaining 40% remain unknown. We also cannot fully understand the experience of burnout for physiotherapists through their survey response. For example, some physiotherapists may leave the workforce due to burnout while some may persevere or use unknown coping strategies to continue in their post. Understanding the experience of burnout would help inform the creation of strategies. It is also unknown if the adaptation of the Likert scales used in the survey impacted the sensitivity or specificity of participant scores. For example, with fewer points on the scale, we may have missed small differences in burnout level between participants. However, it is also unclear how much this other detail would have contributed to our understanding and evaluation of the problem or if it led to a greater prevalence of perfectionism, moral injury, and burnout in the population. It may also be possible that the use of social media in our convenience sampling methods could have introduced 'unknown errors' with the potential to affect accuracy and generalisability, potentially skewing data. Our inclusion criteria cast a wide net, including all HCPC registered physiotherapists in the UK. It is possible other variables, such as a history of depression or other conditions impacting health and well-being may predispose some populations to greater burnout risk. We are unable to determine how these variables might have influenced the data and might be useful to consider for future research on the subject.

Naturally with most surveys, participant answers are subjective, vulnerable to over or underestimation, and only offer us a time snapshot of perfectionism, moral injury, and burnout in UK-based physiotherapists. We cannot understand if these rates are static, vary over the course of the year, or if they still remain high at nearly one year after the start of data collection.

Future research may wish to explore how burnout changes over time through a longitudinal study, with intervention, and across different health professions. Nonetheless, the results may be generalisable to an extent as burnout can be observed in physicians and medical doctors and thus the path of perfectionism-moral injury-burnout may also be present in these populations too [20].

In this publication, we focused on the role of SOP and SPP and their relationship with moral injury and burnout. It may also be of interest to understand the role of OOP. OOP describes behaviours related to the placement of high standards on others. As we know these standards negatively impact physiotherapists with expectations of perfection placed upon them by others, an understanding of the prevalence of OOP may help inform strategies to reduce toxic behaviours in team dynamics and the wider profession. It may also prove informative to explore sleep hygiene practices and sleep quality in the profession to better understand the connection of insomnia with burnout.

## Conclusion

This study offers valuable insights into the intricate dynamics among perfectionism, moral injury, and burnout, emphasising that moral injury acts as a pivotal mediator in the relationship between SPP and burnout. The collective influence of perfectionism and moral injury emerges as a robust predictor of burnout in physiotherapists. Importantly, our findings contribute to the growing body of evidence highlighting the health risks associated with SPP across diverse populations.

The unprecedented levels of burnout observed in UK-based physiotherapists, as revealed by our study, underscore a critical concern with far-reaching implications. This high prevalence of burnout, comparable to a fraying thread in the meticulously woven tapestry of healthcare, has the potential to impact not only the well-being of physiotherapy professionals but also poses risks to patient care, staff dynamics, and the broader society. The implications extend beyond individual practitioners to the healthcare ecosystem as a whole.

In essence, this study highlights the urgent need for targeted and systemic interventions along with comprehensive strategies to address the pervasive challenges posed by perfectionism, moral injury, and burnout within the field of physiotherapy. Acknowledging and actively mitigating these issues is imperative for fostering a resilient and sustainable healthcare workforce, ultimately enhancing the quality of patient care and safeguarding the well-being of both physiotherapists and the communities they serve.

## Supporting information

**S1 Table. STROBE statement checklist.**
(DOCX)

**S1 File. Burnout study data June 2024.**
(XLSX)

## Acknowledgments

This work praises the time and effort spent by the Physiotherapists in the UK.

## Author Contributions

**Conceptualization:** Daniel Biggs, Laura Blackburn, Cameron Black, Sivaramkumar Shanmugam.

**Data curation:** Daniel Biggs, Laura Blackburn, Sivaramkumar Shanmugam.

**Formal analysis:** Daniel Biggs, Laura Blackburn.

**Methodology:** Daniel Biggs, Laura Blackburn, Cameron Black, Sivaramkumar Shanmugam.

**Project administration:** Daniel Biggs, Laura Blackburn, Cameron Black, Sivaramkumar Shanmugam.

**Resources:** Laura Blackburn.

**Software:** Daniel Biggs.

**Supervision:** Sivaramkumar Shanmugam.

**Writing – original draft:** Daniel Biggs, Laura Blackburn.

**Writing – review & editing:** Daniel Biggs, Laura Blackburn, Cameron Black, Sivaramkumar Shanmugam.

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
