## [Decision Letter · Decision Letter 0]

19 Mar 2024

PONE-D-24-04703Physiotherapy Under Pressure: A Cross-Sectional Study on The Interplay Between Perfectionism, Moral Injury, and BurnoutPLOS ONE

Dear Dr. Shanmugam,

Thank you for submitting your manuscript to PLOS ONE. After careful consideration, we feel that it has merit but does not fully meet PLOS ONE’s publication criteria as it currently stands. Therefore, we invite you to submit a revised version of the manuscript that addresses the points raised during the review process.

We look forward to receiving your revised manuscript.

Kind regards,

Renato S. Melo, PhD

Academic Editor

PLOS ONE

3. In this instance it seems there may be acceptable restrictions in place that prevent the public sharing of your minimal data. However, in line with our goal of ensuring long-term data availability to all interested researchers, PLOS’ Data Policy states that authors cannot be the sole named individuals responsible for ensuring data access (http://journals.plos.org/plosone/s/data-availability#loc-acceptable-data-sharing-methods).

Reviewers' comments:

Reviewer's Responses to Questions

**Comments to the Author**

1. Is the manuscript technically sound, and do the data support the conclusions?

Reviewer #1: Yes

Reviewer #2: Partly

2. Has the statistical analysis been performed appropriately and rigorously? 

Reviewer #1: Yes

Reviewer #2: Yes

3. Have the authors made all data underlying the findings in their manuscript fully available?

Reviewer #1: Yes

Reviewer #2: Yes

4. Is the manuscript presented in an intelligible fashion and written in standard English?

Reviewer #1: Yes

Reviewer #2: Yes

5. Review Comments to the Author

Reviewer #1: Abstract

- What is the period analyzed?

- What statistical tests were used?

Text

-The use of the Bayeux Tapestry to exemplify perfectionism did not seem within the context for the work in introduction

The introduction is too long and would need to be succinctly rewritten with the main points.

It is important to bring concepts and approaches directly to the profession As this is a research article on a specific population,

Some information is important, such as the means of inviting participants, the approach, the length of time in the profession considered.

The initial part of the discussion presents results.

Some questions could be relevant and be in the discussion: - how does the profession of physiotherapist differ from its equivalents in the United Kingdom? - what are the relevant laws that could support professional physiotherapists? - how do professional councils deal with this, is there any support for physiotherapists?

Reviewer #2: I would like to extend my gratitude to the authors for undertaking this significant endeavor and commend their valuable contributions to physical therapy research.

Your study addresses an important and timely issue within the field of physiotherapy, exploring the intricate relationships between perfectionism, moral injury, and burnout. The focus on physiotherapists in the UK adds valuable context to the existing literature. The discussions of your findings, especially in context for the NHS has practical real-world relevance. I found your analogy to the "Bayeux Tapestry" to be a creative analogy to tie the topic of perfectionism together in physical therapy.

Overall the manuscript starts with a well structured abstract gives a clear overview of the study objectives, methods, and key findings. The introduction and subsequent sections are comprehensive but there are several areas where improvements can be made for clarity, cohesion and rigor.

Bayeux Tapestry Analogy: The analogy is creative, but the transition between the history of the artifact and the topic of perfectionism could be smoother.

Moral Injury and Burnout sections: These sections are informative and lay the groundwork for the context, but there needs to be a more clear link between the constructs and the physiotherapy profession.

Sampling Method: The use of convenience sampling may limit the generalizability of your findings.

Missing Data: The claim of having no missing data is unusual for a study of this scale and nature. This claim warrants further explanation as missing data is a common issue in survey research

Exclusion Criteria: The lack of specified exclusion criteria may raise concerns about the validity and clarity of your findings.

It is essential for the authors to consider and address the reviewer's feedback in order to improve the clarity, cohesion, and rigor of the manuscript.

6. PLOS authors have the option to publish the peer review history of their article (what does this mean?). If published, this will include your full peer review and any attached files.

Reviewer #1: No

Reviewer #2: **Yes: **Jennifer A Bent

---

## [Author Response · Author response to Decision Letter 0]

7 May 2024

Have included a response to reviewers documents addressing all reviewers and editors comments

---

## [Decision Letter · Decision Letter 1]

24 May 2024

PONE-D-24-04703R1Physiotherapy Under Pressure: A Cross-Sectional Study on The Interplay Between Perfectionism, Moral Injury, and BurnoutPLOS ONE

Dear Dr. Shanmugam,

Thank you for submitting your manuscript to PLOS ONE. After careful consideration, we feel that it has merit but does not fully meet PLOS ONE’s publication criteria as it currently stands. Therefore, we invite you to submit a revised version of the manuscript that addresses the points raised during the review process.

We look forward to receiving your revised manuscript.

Kind regards,

Renato S. Melo, PhD

Academic Editor

PLOS ONE

Journal Requirements:

Reviewers' comments:

Reviewer's Responses to Questions

**Comments to the Author**

1. If the authors have adequately addressed your comments raised in a previous round of review and you feel that this manuscript is now acceptable for publication, you may indicate that here to bypass the “Comments to the Author” section, enter your conflict of interest statement in the “Confidential to Editor” section, and submit your "Accept" recommendation.

Reviewer #1: All comments have been addressed

Reviewer #2: (No Response)

2. Is the manuscript technically sound, and do the data support the conclusions?

Reviewer #1: Yes

Reviewer #2: (No Response)

3. Has the statistical analysis been performed appropriately and rigorously? 

Reviewer #1: Yes

Reviewer #2: Yes

4. Have the authors made all data underlying the findings in their manuscript fully available?

Reviewer #1: Yes

Reviewer #2: Yes

5. Is the manuscript presented in an intelligible fashion and written in standard English?

Reviewer #1: Yes

Reviewer #2: Yes

6. Review Comments to the Author

**Reviewer #1: **Dear authors,

it appears that the revisions were considered. The suggestion is that the conclusion could be more succinct.

**Reviewer #2: **The rewrite is excellent! The introduction flows beautifully, the models are clearly explained and contrasted, and the focus on SPP vs. OPP is well-justified. The sections on moral injury and burnout are much easier to read and understand. Overall, the paper is well-researched, informative, and engaging. Well done!

I only have a few small suggestions that would help reinforce the metaphor to reinforce the theme of precision and complexity of the Bayeux Tapestry that is the central challenge faced by physiotherapists.

They are as follows:

Lines 77-86

Consider Integrating the Bayeux Tapestry into this closing paragraph to help establish a direct comparison between the theme of precision and complexity and the therapist's work.

"This poses a challenge similarly to the embroiders of the Bayeux Tapestry, who meticulously crafted each threat to create a flawless masterpieces. Like the tapestry, physiotherapist must carefully consider numerous factors, meticulously weaving together individual patients needs with the intricate patterns of organizational demands. They are tasked with harmonizing these diverse needs and standards, striving for a tapestry of care that is both comprehensive and precise. However, this intricate balancing act, much like the creation of the tapestry, may introduce elements of unpredictability, potentially leading to discrepancies among the therapist, patient, and the organization, which could ultimately compromise the practioner’s ability to provide optimal care."

Line 90-92 Consider revising " Strict adherence to SOPs, reminiscent of the painstaking precision required to create the Bayeux Tapestry, can leave healthcare professionals vulnerable to heightened exhaustion and cognitive weariness as they strive for perfection

Lines 441-442 Consider adding back a subtle connection and reinforce the message:

This high prevalence of burnout, akin to a fraying thread in the meticulously woven tapestry of healthcare, has the potential to impact not only the well-being of physiotherapy professionals but also poses risks to patient care, staff dynamics, and the broader society.

7. PLOS authors have the option to publish the peer review history of their article (what does this mean?). If published, this will include your full peer review and any attached files.

Reviewer #1: No

Reviewer #2: **Yes: **Jennifer A Bent

---

## [Author Response · Author response to Decision Letter 1]

18 Jun 2024

All suggestions for revisions were actioned with slight modifications to ensure the section flows betters.

---

## [Decision Letter · Decision Letter 2]

2 Jul 2024

Physiotherapy Under Pressure: A Cross-Sectional Study on The Interplay Between Perfectionism, Moral Injury, and Burnout

PONE-D-24-04703R2

Dear Dr. Shanmugam,

We’re pleased to inform you that your manuscript has been judged scientifically suitable for publication and will be formally accepted for publication once it meets all outstanding technical requirements.

Kind regards,

Renato S. Melo, PhD

Academic Editor

PLOS ONE

Additional Editor Comments (optional):

Reviewers' comments:

Reviewer's Responses to Questions

**Comments to the Author**

1. If the authors have adequately addressed your comments raised in a previous round of review and you feel that this manuscript is now acceptable for publication, you may indicate that here to bypass the “Comments to the Author” section, enter your conflict of interest statement in the “Confidential to Editor” section, and submit your "Accept" recommendation.

Reviewer #2: All comments have been addressed

2. Is the manuscript technically sound, and do the data support the conclusions?

Reviewer #2: Yes

3. Has the statistical analysis been performed appropriately and rigorously? 

Reviewer #2: Yes

4. Have the authors made all data underlying the findings in their manuscript fully available?

Reviewer #2: Yes

5. Is the manuscript presented in an intelligible fashion and written in standard English?

Reviewer #2: Yes

6. Review Comments to the Author

Reviewer #2: Dear Authors:

Your revised manuscript is a significant improvement and a testament to the tremendous work and dedication you have all put into it. The revised introduction and sections on alternative models is excellent. The moral injury and burnout sections along with the Shirom Melamed Burnout Measure helps to highlight the importance of assessing and addressing burnout in the the profession. Finally, the refined metaphors, carefully woven within your own tapestry now beautifully reinforces the central theme.

Overall, the changes you have implemented have greatly elevated this piece which makes it a valuable contribution to field. You have satisfied my concerns and I have no other comments other than to offer my congratulations and admiration for a job well done!

Cheers to you all! Enjoy a drink together!

7. PLOS authors have the option to publish the peer review history of their article (what does this mean?). If published, this will include your full peer review and any attached files.

Reviewer #2: **Yes: **Jennifer A. Bent

---

## [Editor Report · Acceptance letter]

9 Jul 2024

PONE-D-24-04703R2 

PLOS ONE

Dear Dr. Shanmugam, 

I'm pleased to inform you that your manuscript has been deemed suitable for publication in PLOS ONE. Congratulations! Your manuscript is now being handed over to our production team.

Kind regards, 

on behalf of

Dr. Renato S. Melo 

Academic Editor

PLOS ONE